# PROXYATTN: GUIDED SPARSE ATTENTION VIA REPRESENTATIVE HEADS

**Yixuan Wang**[1,*]  **Huang He**[2]  **Siqi Bao**[2]  **Hua Wu**[2]  **Haifeng Wang**[2]
**Qingfu Zhu**[1]  **Wanxiang Che**[1,†]

[1]Harbin Institute of Technology, China
[2]Baidu Inc., Beijing, China
`{yixuanwang, qfzhu, car}@ir.hit.edu.cn`
`{hehuang, baosiqi, wu_hua, wanghaifeng}@baidu.com`

## ABSTRACT

The quadratic complexity of attention mechanisms limits the efficiency of Large Language Models (LLMs) on long-text tasks. Recently, methods that dynamically estimate block importance have enabled efficient block sparse attention, leading to significant acceleration in long-text pre-filling of LLMs. However, their coarse-grained estimation inevitably leads to performance degradation at high sparsity rates. In this work, we propose *ProxyAttn*, a training-free sparse attention algorithm that achieves more precise block estimation by compressing the dimension of attention heads. Based on our observation of the similarity among multiple attention heads, we use the scores of pooled representative heads to approximate the scores for all heads. To account for the varying sparsity among heads, we also propose a block-aware dynamic budget estimation method. By combining the scores from representative proxy heads with multi-head dynamic budgets, we achieve a more fine-grained block importance evaluation at low computational cost. Experiments on a variety of mainstream models and extensive benchmarks confirm the underlying similarity among attention heads. Leveraging a fine-grained estimation, the proposed method achieves substantial gains in performance and efficiency compared to existing methods. More precisely, ProxyAttn can achieve up to 10.3x attention acceleration and 2.4x prefilling acceleration without significant performance loss. Our code is available at  https://github.com/wyxstriker/ProxyAttn.

## 1 INTRODUCTION

Large Language Models (LLMs) have shown outstanding performance in long context tasks (Liu et al., 2025a;b) and are widely applied in fields such as code generation (Jimenez et al., 2023; Hui et al., 2024) and mathematical reasoning (Chen et al., 2025; Guo et al., 2025). However, transformer-based LLMs face efficiency limitations when processing long texts with millions of tokens. The quadratic complexity of the attention mechanism poses a significant challenge to the inference of models. Recently, numerous studies (Brauwers & Frasincar, 2021; Zhang et al., 2025) have worked on refining the attention module to increase the training and inference efficiency.

In particular, sparse attention methods mitigate the latency issues of the attention mechanism by skipping certain computations. By leveraging the inherent sparsity (Liu et al., 2022; Deng et al., 2024) of attention score distributions, these methods can reduce computational costs while preserving model performance. Some static methods (Beltagy et al., 2020; Zaheer et al., 2020) use heuristic fixed templates to achieve sparse computation, making it difficult to maintain model performance with diverse attention patterns (Likhosherstov et al., 2021). To achieve a more accurate estimation, many efforts have focused on exploring various low-cost method to assess the importance of tokens, which in turn enables the dynamic sparse patterns. The performance of hardware-friendly sparse

---

*Work done during internship at Baidu, Inc.
†Corresponding author.

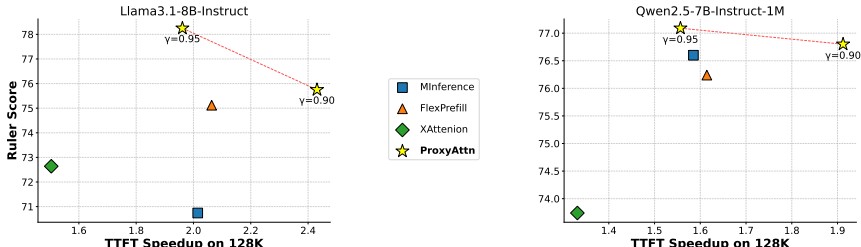

Figure 1: Performance and speedup results of different sparse attention methods on RULER.

attention, which is typically handled at the block level (Guo et al., 2024), is heavily dependent on the efficiency and accuracy of its importance estimation. MInference (Jiang et al., 2024) leverages multiple pre-defined templates and an online computed index to guide the construction of block sparse patterns. Furthermore, FlexPrefill (Lai et al., 2025) proposes an input-aware approach that dynamically determines the sparse patterns and sparsity rates.

While existing methods have proven effective in long-context scenarios, their performance at high sparsity rates is still **limited by their coarse-grained importance estimation**. To evaluate the importance of each block, these methods typically compress along the sequence dimension to approximate their attention scores. This coarse-grained compression can cause a few high-scoring positions to be overlooked, which in turn impacts the final performance. A token-level dot-product calculation would solve this issue but is not feasible, as its time complexity scales identically to full attention.

In this paper, we achieve a fine-grained and efficient trade-off in block importance estimation by exploring compression of the attention head dimension. We begin by analyzing the performance of multi-head attention on long sequences, where we observed two primary characteristics of how the heads behave: **(1) The overall attention trends of different heads are consistent. (2) The primary differences between heads lie in their sparsity.** Based on these findings, we propose *ProxyAttn*, a training-free sparse attention algorithm that approximates attention scores using a few representative heads. Specifically, we leverage the similarity among attention heads to approximate the full attention scores using the complete scores from a few pooled heads. By incorporating max pooling, our proposed method enables more accurate block importance estimation while ensuring efficiency. Furthermore, given the significant differences in head sparsity, we propose an online block-aware budget estimation method. By integrating the unified importance score with head-level budget, we can achieve diverse sparse attention patterns across different heads.

In order to evaluate our proposed method, we conduct a series of extensive experiments. We use widely-adopted LLMs that are specifically designed for long-context tasks like Llama3.1-8B-Instruct (Grattafiori et al., 2024) and Qwen2.5-7B-Instruct-1M (Yang et al., 2025), along with a variety of mainstream benchmarks such as RULER (Hsieh et al., 2024), InfiniteBench (Zhang et al., 2024), and LongBench-v2 (Bai et al., 2024a;b). Experimental results show that ProxyAttn significantly outperforms other methods on both synthetic and real-world datasets. As illustrated in Figure 1, the proposed ProxyAttn achieves a Pareto front for performance and efficiency compared to other coarse-grained estimation methods. Our contributions can be summarized as follows:

- We introduce ProxyAttn, a training-free sparse attention estimation method. It leverages the attention scores from a small number of proxy heads to efficiently estimate the scores for all heads, achieving more accurate importance measures.

- Considering the diverse sparsity among heads, we propose a block-aware budget allocation method. Coupled with the unified importance score, our method can provide different head sparsity budgets online, which in turn yields diverse sparse attention masks.

- Extensive experiments demonstrate that the proposed ProxyAttn outperforms other coarse-grained sparse attention methods across various model architectures and benchmarks. Our approach yields a maximum 10.3x speedup for attention computation without significant performance loss.

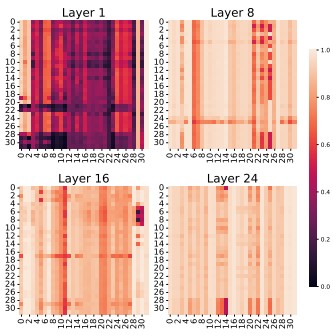 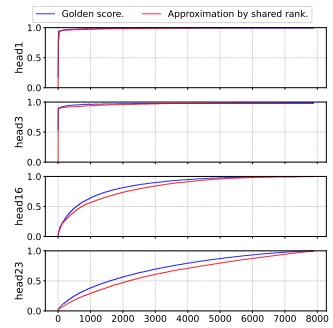 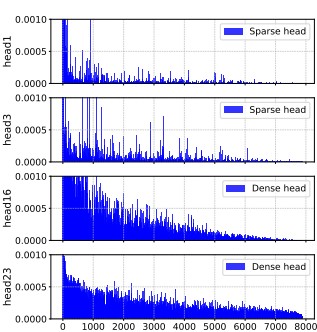

(a) The tokens that attention heads focus on exhibit a consistency in long context.

(b) A shared token ranking can simultaneously accommodate the scores from different heads.

(c) Attention heads vary in their sparsity. Both sparse and dense heads still share a common focus on certain tokens.

Figure 2: Observational study on Llama3.1-8B-Instruct using 8K-token data from the Needle in a Haystack (NIAH) Single-1 task of the RULER (Hsieh et al., 2024) dataset.

## 2 OBSERVATION

### 2.1 MOTIVATION

Recent work (Jiang et al., 2024; Lai et al., 2025; Gao et al., 2024) on the importance of different attention blocks typically employs heuristic "vertical and slash" patterns or pooling methods along the sequence dimension for approximation. These methods either rely on specific heuristic template assumptions or opt for a coarse-grained estimation, rendering the importance measures of blocks inaccurate (Xu et al., 2025). Given the inherent sparsity of attention distributions, such coarse-grained pooling can easily overlook a few high-scoring positions.

A viable approach for both accuracy and efficiency in token-level dot-product computations is to compress along the head num dimension instead of the sequence dimension. This allows a few key heads to serve as proxies for estimating the importance of all heads. However, the feasibility of this approach hinges on the assumption that attention heads exhibit a certain degree of consistency in their scores. To validate this hypothesis, we conduct some observational studies on the behaviors of multiple heads in a long-context setting.

### 2.2 CHARACTERISTICS OF MULTIPLE ATTENTION HEADS

To investigate the similarities and differences among different heads, we conduct experiments on the Llama model using long-context data containing 8K tokens. As shown in Figure 2, by observing the performance of different heads, we can derive two key findings:

**Attention heads exhibit consistency in token focus.** We first calculate the overlap of tokens that different heads primarily focus on. As shown in Figure 2a, the score at position (i, j) represents the cumulative attention score at head $j$ obtained by the top 1024 tokens from head $i$. It can be observed that while the very dense first layer has inherently low recall, the attention overlap scores between different heads in other layers are very high. This suggests that in long texts, multiple attention heads, particularly in deeper layers, focus on a large intersection of tokens.

To further assess this similarity, we consider whether it could be captured by a shared metric, with a natural candidate being average pooling across the heads. As depicted in Figure 2b, we sort the tokens by their average score across all heads and then analyze the discrepancy between the cumulative probability curve based on a shared ranking and the actual score curve for specific heads. The consistency between the two curves serves as the basis for our ProxyAttn algorithm, which leverages the scores from a few heads to approximate the attention of all heads.

**Attention heads exhibit variability in sparsity.** While the diversity among attention heads is a well-established observation (Jiang et al., 2024), we find that it does not contradict our findings

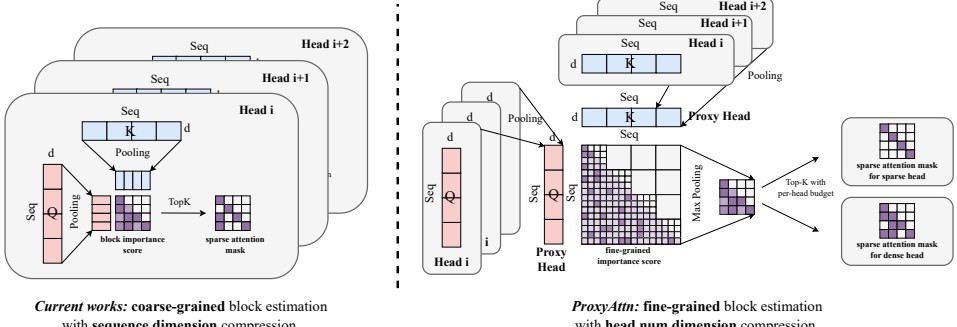

Figure 3: **Illustration of ProxyAttn.** By compressing the head dimension, ProxyAttn can obtain token-level importance scores, leading to more accurate block importance estimation. A proxy head is able to obtain diverse masks by leveraging the online budget estimations from different heads.

of consistency. Consistent with the preceding setup, we utilize the same shared token ranking to examine the attention scores across various heads. Figure 2c illustrates that the primary variation between heads is in their sparsity rather than the tokens they focus on. This observation corroborates the findings of Xiao et al. (2024); Fu et al. (2024). As seen in the figure, given the same ranking, all heads assign high attention to the leading tokens. The main distinction is that some heads are very sparse and only attend to the initial portion, while others are dense and maintain significant scores for later tokens. This motivates us to realize that while leveraging shared scores, a flexible budget must be allocated to different heads to ensure performance.

## 3 PROXYATTN

Building upon our above observations of attention heads, we propose the ProxyAttn algorithm to efficiently achieve accurate, fine-grained importance estimation for block sparse attention. A schematic comparison with existing coarse-grained estimation methods is shown in Figure 3. Leveraging the consistency of token focus among heads, we first employ an intra-group score sharing method (§3.1) for importance estimation. To account for the varying sparsity of heads, we then dynamically assign a unique budget (§3.2) to each head to generate diverse sparse masks.

### 3.1 UNIFIED SCORE ESTIMATION

In order to obtain the token-level dot-product scores for each block in the attention map, compression must be applied along dimensions other than the sequence length to ensure efficiency. Given the consistency of token attention across heads on long contexts, we propose to perform compression along the head num dimension. Specifically, we partition the attention heads into distinct groups. Within each group, a designated head is used to compute attention, which then serves as a proxy to guide the attention scores for the entire group. The entire process can be formulated as:

$$\mathbf{A}^i = \text{maxpool}\left(\text{softmax}\left(\frac{\mathbf{Q}^g \mathbf{K}^{gT}}{\sqrt{d_k}}\right)\right), \quad i \in G_g \quad (1)$$

where $\mathbf{Q}^g, \mathbf{K}^g \in \mathbb{R}^{N \times d_k}$ and $\mathbf{A}_i \in \mathbb{R}^{\frac{N}{b} \times \frac{N}{b}}$. For all heads belonging to group $g$, a single token-level score will be shared. Combined with max-pooling, it enables fine-grained block importance estimation at the cost of computing only $|G|$ proxy heads.

For the representative head queries $\mathbf{Q}^g$ and keys $\mathbf{K}^g$, we heuristically apply the same average pooling as for the sequence. This can also be seen as a form of compression along the head number dimension and can be formalized as:

$$\mathbf{Q}^g = \frac{1}{|G|} \sum_{i \in G} \mathbf{Q}^i \quad \mathbf{K}^g = \frac{1}{|G|} \sum_{i \in G} \mathbf{K}^i \quad (2)$$

It should be noted that for models using Group-Query Attention (GQA) (Ainslie et al., 2023), the group granularity will be aligned with the keys, ensuring that all queries belonging to the same group of keys are also within that same group.

---

**Algorithm 1** Block-aware Budget Estimation for Head $i$

---

**Input:** Query states $\mathbf{Q}^i$, key states $\mathbf{K}^i$, cumulative probability threshold $\gamma$
**Output:** Estimated budget $b_i$

$\quad \hat{\mathbf{A}}^i \leftarrow \text{softmax}\left( \mathbf{Q}^i_{\text{last}} \mathbf{K}^{i\top} / \sqrt{d_k} \right)$ $\qquad\qquad$ ▷ Compute approximate attention scores

$\quad \mathbf{a}^i \leftarrow \text{avgpool}(\hat{\mathbf{A}}^i)$ $\qquad\qquad\qquad\qquad$ ▷ Aggregate into block-level scores

$\quad \mathbf{a}^i \leftarrow \text{sort}(\mathbf{a}^i) / \text{sum}(\mathbf{a}^i)$ $\qquad\qquad\quad$ ▷ Normalize and sort block-level scores

$\quad b_i \leftarrow \min\left\{ k \; \Big| \; \sum_{j=0}^{k} \mathbf{a}^i[j] \geq \gamma \right\} / |\mathbf{a}^i|$ $\qquad$ ▷ Determine budget ratio for selected head

$\quad$ return $b_i$

---

To further reduce the computational cost of the estimation operation, we also dropped certain qk pairs by stride, keeping only the first token in each stride window. Driven by the local similarity inherent in attention mechanisms (Wu et al., 2024), this operation reduces the computational cost without significant loss of accuracy (see Appendix B.2 for a detailed analysis). For a model with $n$ heads, the theoretical computational cost of using $g$ representative proxy heads for estimation is expected to be $\frac{g}{n \cdot \text{stride}^2}$ of full multi-head attention. Leveraging head sharing, the proposed method reduces the computational cost for a 7B model to roughly one percent of full attention under conventional settings. This allows for fine-grained estimation without compromising computational efficiency.

## 3.2 DYNAMIC BUDGET ALLOCATION

After obtaining importance scores for different blocks, a common practice is to further obtain a sparse block mask (Gao et al., 2024; Xu et al., 2025), often by using methods like Top-K selection. However, directly applying this approach to proposed method would result in identical masks for all heads in the same group, thereby impacting the maximum achievable sparsity rate. Given the findings from §2.2, the attention focuses of different heads are consistent, with the primary difference being in their sparsity. We consider an online approach to approximately evaluate the sparsity of individual heads, which in turn guides the selection of the budget.

Inspired by the work of Jiang et al. (2024), we propose to use the queries from the last block to estimate the sparsity of different heads. The budget required for a given head is approximated as the minimum budget for the query of the final block to achieve a cumulative probability of $\gamma$. Furthermore, given the significant discrepancy between token-level computation and the actual block budget due to attention sparsity, we employ average pooling to ensure that the budget estimation is performed at the block level. Algorithm 1 presents the overall approach to budget estimation. After obtaining the budget $\mathbf{b}_i$ for each head, we select the top $b_i$ blocks with the highest unified scores to form the sparse attention masks. The final sparse attention mask $\mathbf{S}^i_{M \times N}$ for head $i$ can be formalized as:

$$\mathbf{S}^i_{mn} = \begin{cases} 1, & \text{if } n \in \text{TopK}(\mathbf{A}^i[m], \mathbf{K} = b_i N).\text{index} \\ 0, & \text{otherwise} \end{cases} \qquad (3)$$

Where $\mathbf{M}, \mathbf{N}$ is the block-level sequence length, and $\mathbf{A}^i$ is the previously estimated block-level importance of head $i$. As shown in Figure 3, the proposed ProxyAttn achieves diverse sparse attention masks by combining shared head scores and the dynamic budget allocation method. By leveraging the efficient block sparse attention kernel (Guo et al., 2024), we can substantially accelerate the attention operation for long contexts while maintaining comparable model performance.

## 4 EXPERIMENTS

### 4.1 EXPERIMENTAL SETUP

**Evaluation datasets.** To thoroughly evaluate the performance of proposed method for long contexts, we utilize two types of benchmarks: synthetic and real-world datasets. For synthetic benchmarks, we adopt the *RULER* (Hsieh et al., 2024), which comprises a variety of synthetic tasks with token lengths ranging from 4K to 128K. This provides a direct way to evaluate the long-context

Table 1: Experimental results on RULER. **Bolded** values represent the best result under the same setting, while underlined values indicate the second best. ∗ denotes methods that require additional training. Sparsity is calculated by weighting the ratio of different lengths by their token count.

| Method | Sparsity↑ | 4k | 8k | 16k | 32k | 64k | 128k | Avg. | wAvg. |
|---|---|---|---|---|---|---|---|---|---|
| *Llama3.1-8B-Instruct* | | | | | | | | | |
| FullAttention | 0.00 | 96.17 | 94.21 | 93.72 | 89.56 | 86.13 | 76.95 | 89.46 | 86.49 |
| Minference | 0.70 | **96.23** | 94.22 | 93.92 | 88.82 | 84.64 | 70.74 | 88.09 | 84.25 |
| FlexPrefill | 0.72 | 96.03 | 94.07 | **94.35** | **92.05** | 85.21 | 75.12 | 89.47 | 86.28 |
| XAttention | 0.69 | 96.19 | 94.27 | 93.99 | 91.04 | 85.15 | 72.64 | 88.88 | 85.35 |
| SeerAttention∗ | 0.77 | 96.20 | 94.44 | 93.80 | 90.01 | 84.99 | 74.34 | 88.96 | 85.60 |
| ProxyAttn ($\gamma$=0.90) | **0.80** | 95.76 | 94.32 | 93.52 | 91.05 | 86.14 | 75.75 | 89.42 | 86.31 |
| ProxyAttn ($\gamma$=0.95) | 0.69 | 96.07 | **94.57** | 94.01 | 91.55 | **86.63** | **78.25** | **90.18** | **87.43** |
| *Qwen2.5-7B-Instruct-1M* | | | | | | | | | |
| FullAttention | 0.00 | 94.62 | 91.45 | 89.49 | 88.46 | 84.58 | 78.90 | 87.92 | 85.53 |
| Minference | 0.63 | 94.41 | 91.43 | 89.18 | 87.56 | 82.73 | 76.60 | 86.99 | 84.20 |
| FlexPrefill | 0.69 | 94.09 | 90.97 | 89.27 | **87.59** | 81.85 | 76.24 | 86.67 | 83.85 |
| XAttention | 0.61 | 92.92 | 88.22 | 84.45 | 84.09 | 79.99 | 73.74 | 83.90 | 81.02 |
| ProxyAttn ($\gamma$=0.90) | **0.74** | 94.57 | 91.34 | 89.19 | 86.55 | **83.76** | 76.80 | 87.04 | 84.32 |
| ProxyAttn ($\gamma$=0.95) | 0.61 | **94.61** | **92.07** | **89.75** | 86.68 | 83.57 | **77.09** | **87.30** | **84.53** |

capabilities of models and the performance trade-off introduced by sparse attention. Besides, We use *InfiniteBench* (Zhang et al., 2024) and *LongBench-v2* (Bai et al., 2024a;b) as benchmarks for real-world tasks. Both include diverse long-text tasks, such as QA and summarization, providing a more comprehensive evaluation in practical applications.

**Selected baselines.** To fairly evaluate the proposed *ProxyAttn*, we compare it against the following mainstream block sparse attention methods as strong baselines: (1) *MInference* (Jiang et al., 2024) enables dynamic sparse attention by defining three unique patterns and estimating the optimal indices for each pattern at inference time. (2) *FlexPrefill* (Lai et al., 2025) leverages input statistics to enable context-aware sparse pattern determination and Top-P block selection, making the application of sparse attention more adaptable. (3) *XAttention* (Xu et al., 2025) mitigates the neglect of important tokens after pooling by more effectively capturing the Vertical-Slash pattern through the computation of the anti-diagonal. (4) *SeerAttention* (Gao et al., 2024) employs a trainable MLP to process the latent information from the pooled queries and keys, which allows it to capture block importance more effectively than direct pooling.

For the hyperparameter settings of the baseline methods, we strictly follow the optimal parameters reported in their original papers. Specifically, MInference pre-configures patterns and budgets for each head. To ensure fair sparsity, we configure FlexPrefill with a $\gamma$ of 0.95 and a minimum budget of 2048. While XAttention employs head budgets determined via offline search, for models like Qwen where these budgets are not provided, we directly set a threshold of 0.95 to ensure the method's performance. SeerAttention implements dynamic block sparsity based on a threshold, which we set to 5e-4 as original paper.

**Implementation details.** Consistent with previous studies Jiang et al. (2024); Lai et al. (2025), we choose to evaluate sparse attention methods exclusively during the **pre-filling** stage, which is the primary computational bottleneck, while using full attention for the decoding stage. All performance and speed experiments are conducted on an NVIDIA-H800-80GB platform. We fuse dot-product and max-pooling operators to reduce latency during the block estimation phase, following the kernel design of (Gao et al., 2024). For the subsequent block sparse attention stage, we leverage the efficient implementation proposed by Guo et al. (2024).

For the hyperparameter selection of our proposed ProxyAttn, we adopt two settings with $\gamma$ values of 0.95 and 0.90, respectively, while maintaining a stride of 4 and a minimum budget of 2048 tokens. See Appendix B for more on hyperparameter experiments. When choosing the number of proxy heads for estimation, we selected 1 and 4 for the LLaMA and Qwen models, considering the similarity of their attention heads. This indicates that a calculation using just one head accurately represent the behavior of all 32 heads in the LLaMA3.1-8B model. We will discuss this further in Section 5.1.

Table 2: Main Results on InfiniteBench and LongBench-v2. LongBench results are reported without Chain-of-Thought (CoT), with the "w. CoT" results shown in parentheses. The overall score is obtained by averaging the final scores from the two benchmarks.

| Methods | InfiniteBench | | | | | LongBench-v2 (w. CoT) | | | Overall |
|---|---|---|---|---|---|---|---|---|---|
| | En | Zh | Code | Math | Retri. | Short | Medium | Long | |
| *Llama3.1-8B-Instruct* | | | | | | | | | |
| FullAttention | 32.09 | 13.65 | 23.35 | 33.43 | 84.84 | 33.90 (36.10) | 26.50 (30.70) | 25.00 (27.80) | 35.38 |
| MInference | 29.00 | 12.57 | **26.40** | **35.71** | 73.59 | **35.00** (38.90) | 26.00 (**31.20**) | **30.60** (26.90) | 34.78 |
| FlexPrefill | 30.75 | **13.69** | 23.60 | 32.29 | 80.77 | 30.60 (36.70) | **27.90** (25.10) | 25.90 (28.70) | 33.96 |
| XAttention | 31.45 | 13.28 | 22.84 | 32.57 | 80.03 | **35.00** (35.00) | 26.00 (**31.20**) | 26.90 (27.80) | 34.89 |
| ProxyAttn | **33.21** | 13.35 | 25.13 | 32.00 | **81.49** | 34.40 (**41.10**) | 27.40 (27.90) | 29.60 (**29.60**) | **36.06** |
| *Qwen2.5-7B-Instruct-1M* | | | | | | | | | |
| FullAttention | 30.57 | 15.65 | 31.98 | 41.14 | 77.87 | 39.40 (38.90) | 28.40 (36.30) | 34.30 (33.30) | 38.22 |
| MInference | 30.12 | 15.16 | 30.20 | **47.71** | **78.53** | 40.60 (**43.90**) | 26.50 (34.40) | 28.70 (29.60) | 37.91 |
| FlexPrefill | **31.42** | **15.66** | 30.96 | 42.29 | 75.07 | 41.70 (35.60) | **30.20** (34.90) | 27.80 (31.50) | 37.39 |
| XAttention | 29.63 | 15.56 | 32.74 | 43.71 | 70.40 | 41.70 (42.20) | 27.40 (33.50) | **32.40** (30.60) | 37.26 |
| ProxyAttn | 29.75 | 15.36 | **32.99** | 41.71 | 77.60 | **43.90** (40.00) | 28.40 (**36.70**) | 27.80 (**32.40**) | **38.33** |

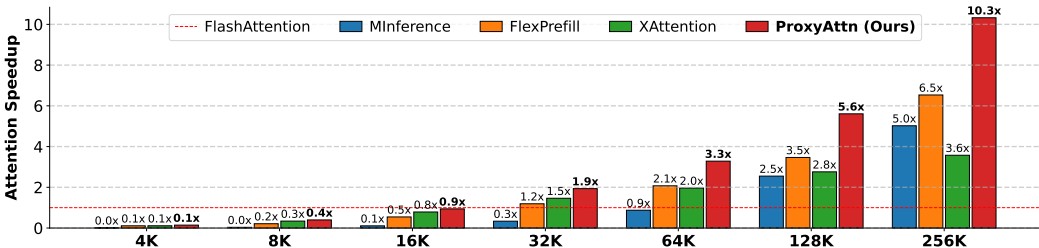

Figure 4: Kernel-level speedup of existing sparse attention methods with varying input lengths.

## 4.2 ACCURACY RESULTS

**Results on synthetic tasks.** Following the settings of Hsieh et al. (2024), we also report weighted average scores (wAvg.), which assigns linear weights to the results based on actual token lengths. The results for synthesis tasks with various length are presented in Table 1, which shows that the proposed method has significant advantages over existing methods in terms of both sparsity and performance. By sharing scores between attention heads, ProxyAttn achieves excellent results on models with varying GQA settings, including Llama3.1 and Qwen2.5. This highlights the strong generalization capabilities of the proposed method.

Additionally, we evaluate the performance of the proposed method at various sparsity rates by adjusting $\gamma$. When ProxyAttn maintains a sparsity ratesimilar to other methods, it achieves the best average performance on RULER, even surpassing the performance of full attention on the Llama model. As the sparsity further increases, the proposed method can still achieve comparable experimental results while also reducing inference latency. In contrast to the performance degradation XAttention faces when transferred to Qwen, ProxyAttn can be quickly migrated to other models without hyperparameter searching by using online budget estimation.

**Results on real-world tasks.** In addition to synthetic tasks, we also evaluate the performance of existing methods on real-world tasks using benchmarks such as InfiniteBench Zhang et al. (2024) and LongBench-v2 Bai et al. (2024b). As shown in Table 2, despite the lower discriminability of real-world tasks compared to synthetic ones, the proposed ProxyAttn still achieve the best average results and even surpassed the performance of full attention. This demonstrates the potential of sparse attention methods for handling long documents. Full experimental results for the two datasets are provided in Appendix C.

## 4.3 EFFICIENCY RESULTS

**Attention speedup.** To intuitively demonstrate the efficiency of our proposed method, we first evaluate its attention acceleration ratio compared to other methods at kernel level. Considering that

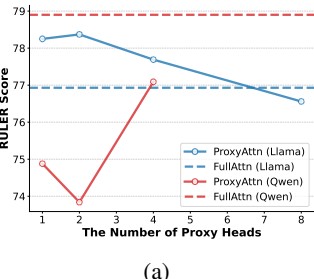 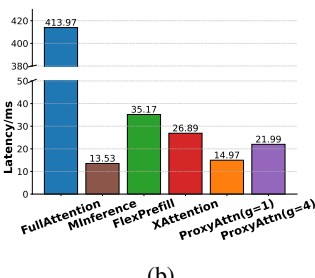 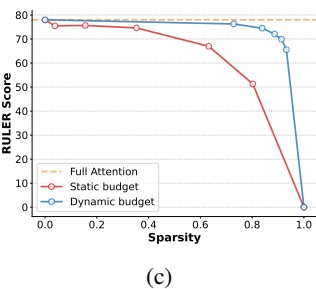

(a)            (b)            (c)

Figure 5: Experimental analysis of proposed method. (a) The performance of different numbers of proxy attention heads across various models. (b) Latency for estimating block importance with 128K inputs using different methods. The latency of all methods is less than 10% of the Full Attention. (c) Performance degradation with increasing sparsity rate for different budget allocation methods.

the inductive bias of sparse attention is not effective with random vectors, we cache the queries, keys, and values with a true distribution from the RULER dataset for speed measurement. The final speedup is obtained by averaging multiple runs across different layers. We use the official implementation of FlashAttention (Dao et al., 2022; Dao, 2024) as our baseline for testing, and the experimental results are shown in Figure 4.

As shown in the figure, sparse attention begins to show its speedup effect when the input text exceeds 32K tokens. Thanks to its fine-grained and low-cost block importance estimation, ProxyAttn can achieve a lower sparsity (see Appendix D) while maintaining performance, thereby leading to a more significant speedup. It is worth noting that the proposed method can achieve a 10.3x attention acceleration with 256K tokens, which would significantly improve the pre-filling speed of the model.

**End-to-end prefilling speedup.** The pre-fill latency of LLMs is also influenced by the MLP modules, which diminishes the acceleration effect gained from optimizing the attention operation alone. As shown in Figure 1, we report the performance and Time to First Token (TTFT) speedup of our proposed method under the RULER 128K setting. By adjusting the settings, the proposed Proxy-Attn achieves a Pareto frontier of performance and efficiency with its fine-grained block importance estimation. With a minimal loss of model performance, an end-to-end speedup of up to 2.4x is achieved.

## 5 ANALYSIS

### 5.1 ATTENTION HEAD SIMILARITIES ACROSS MODELS

The effectiveness of ProxyAttn is grounded in the existence of similarities among different heads. To compare how this phenomenon varies across different models, we conduct experiments on RULER 128K with a varying number of proxy heads. Figure 5a shows the results where we test models with different GQA configurations by varying the number of proxy heads from 1 to the number of key heads.

The multiple attention heads of the Llama model exhibit highly consistent performance. Consequently, a single proxy head can effectively represent all 32 heads, achieving performance that surpasses the baseline. Furthermore, for the Qwen model, approximately 4 proxy heads are typically required to achieve satisfactory performance. We hypothesize that this may be related to its very high GQA grouping ratio (seven queries per key). Although increasing the number of proxy heads raises the latency of importance estimation, the overall latency is not significant due to the use of strided dropout on the sequence dimension.

### 5.2 COMPARISON OF ESTIMATION LATENCY

Beyond achieving a higher sparsity rate, the inherent latency of the block estimation algorithm must also be factored into the overall algorithmic performance. To this end, we evaluate the overhead of existing sparse attention methods at the block estimation stage. As shown in Figure 5b, all methods achieve an estimation latency of less than 10% of that of full attention. In particular,

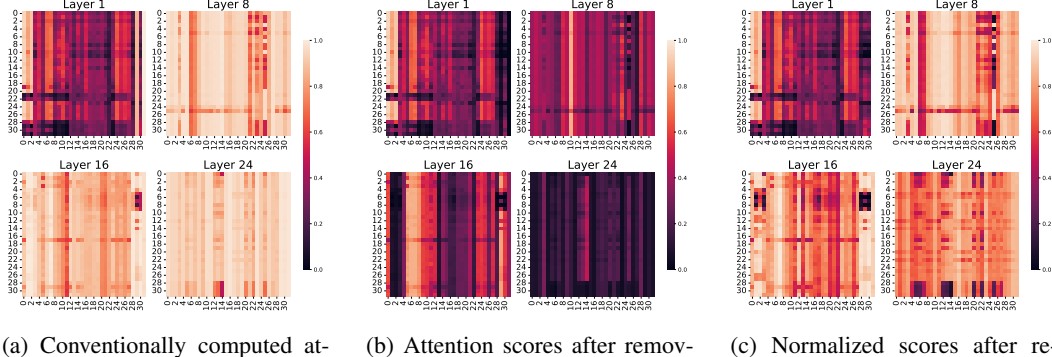

(a) Conventionally computed attention scores.

(b) Attention scores after removing sink tokens (n=4).

(c) Normalized scores after removing sink tokens (n=4).

Figure 6: Illustration of token overlap across different heads.

the proposed ProxyAttn reduces computational costs by compressing the attention head dimension, thereby allowing it to account for more fine-grained, token-level dot-products while maintaining a latency comparable to other methods. Furthermore, due to the use of dropout with a small stride, increasing the number of proxy heads does not significantly increase the estimation latency, which further enhances the generalizability of the method.

### 5.3 THE EFFECT OF DYNAMIC BUDGET ALLOCATION

As discussed in §2.2, the sparsity among attention heads is highly variable. Consequently, if a static Top-K allocation method is employed, it can lead to excessive information loss in some dense heads, which in turn impacts the final performance. To validate our hypothesis, we compare the performance degradation of static and dynamic methods as the sparsity rate increased, using 128K-token inputs. As shown in Figure 5c, the proposed method significantly outperforms the static baseline, exhibiting a notable performance degradation only when the sparsity exceeds 90%. This highlights the importance of independently estimating the budget for each head.

### 5.4 MODEL SCALING EXPERIMENTS

Given that larger models typically have more attention heads, we scale our experiments to a 70B model to validate the effectiveness of our method with a greater number of attention heads. To ensure a fair comparison, we also conduct experiments on several baselines that are compatible with the 70B model, while maintaining the same configurations as primary experiments. As shown in Table 3, even with the number of attention heads increased to 64,

Table 3: Ruler results on Llama3.1-70B-Instruct with 64 attention heads.

| Method | RULER | TTFT/s |
|---|---|---|
| FullAttention | 65.03 | 91.84 (1.00×) |
| MInference | 60.33 | 82.90 (1.11×) |
| XAttention | 58.59 | 60.57 (1.52×) |
| ProxyAttn | **62.23** | **44.23 (2.08×)** |

we can still achieve effective block importance estimation using a single proxy head. While maintaining optimal performance, we can achieve a 2.08x end-to-end prefilling speedup for Llama3.1-70B on 128K sequences.

### 5.5 EFFECT OF SINK TOKENS

Recent studies (Xiao et al., 2023) have found that sink tokens, typically located at the beginning of a sequence, often receive a large amount of attention score. To eliminate the influence of sink tokens, we conduct a revised observational experiment based on §2.2, removing attention scores to the sink tokens. For a given token, let its attention scores be $\mathcal{A}$, with shape (`head_num, seq_len`). The overlap scores at (`i, j`) in the figure are computed as follows:

$$\text{token\_index} = \text{topk}(\mathcal{A}[i], k = 1024).\text{indices}$$
$$\text{Score}[i, j] = \text{sum}(\mathcal{A}[j, \text{token\_index}]) \tag{4}$$

The experimental results are shown in Figure 6. It can be observed that as the layer depth increases, the attention scores received by sink tokens grow progressively larger. After re-normalizing the attention, there is still substantial overlap across different heads. This indicates that the improvements provided by the proxy head are not merely a result of shared sink tokens, but also reflect its capacity to act as a proxy at the global level.

## 6 RELATED WORK

**Multi-head attention.** While offering powerful modeling capabilities, Multi-Head Attention (MHA) (Vaswani et al., 2017) inherently suffers from efficiency issues. Many recent studies (Zheng et al., 2024) aim to refine the attention mechanism from the perspective of multiple heads. Multi-Query Attention (MQA) (Shazeer, 2019) and Grouped-Query Attention (GQA) (Ainslie et al., 2023) significantly reduce the memory footprint of the KV cache by having multiple queries share a single set of keys and values. Multi-head Latent Attention (MLA) (Liu et al., 2024) implicitly groups via low-rank projection, achieving a further reduction in KV Cache memory consumption.

In addition, other work investigates the differential performance across various heads. MoA (Fu et al., 2024) achieves efficient attention computation by assigning diverse, sparse attention patterns to each head in every layer. DuoAttention (Xiao et al., 2024) utilizes an optimization method to effectively identify sliding-window heads and subsequently performs the corresponding sparse computation. Beyond the sparsity differences discussed above, the findings of this paper also indicate a potential consistency across multiple heads, which can be leveraged for designing subsequent efficient attention architectures.

**Sparse attention.** Exploiting the inherent sparsity of the attention mechanism, MInference (Jiang et al., 2024) and FlexPrefill (Lai et al., 2025) achieve efficient block-sparse attention by computing a dynamic sparse mask. SeerAttention (Gao et al., 2024) introduces a trainable MLP to extract information from different blocks, leading to a more accurate estimation. XAttention extends the pooling-based methods by incorporating a more fine-grained anti-diagonal scoring mechanism, which leads to a better capture of the heuristic "vertical and slash" patterns. Similarly inspired by inter-head similarity, SharePrefill (Peng et al., 2025) performs offline clustering of attention heads. By fully sharing sparse patterns among similar heads, it achieves more accurate block attention estimation.

However, directly applying sparse attention to LLMs still incurs an unavoidable performance degradation. Some approaches (Team et al., 2025) attempt to introduce optimization objectives during training to induce native sparsity of LLMs. NAS (Yuan et al., 2025) achieves a native sparse attention that surpasses full attention by integrating three modes: compression, selection, and sliding window. Referencing the principles of MoE (Mixture-of-Experts), MOBA (Lu et al., 2025) achieves a mix of block sparse attention without introducing any additional parameters.

## 7 CONCLUSION

In this paper, we propose ProxyAttn, a training-free method for sparse attention. Considering the similarity among multiple attention heads, we leverage proxy heads to efficiently estimate the attention scores for all heads. When integrated with dynamic budget allocation, ProxyAttn outperforms existing approaches across multiple datasets and models. We will further investigate the utility of the proxy head specifically within the decoding phase of LLMs.

## ACKNOWLEDGEMENTS

We gratefully acknowledge the support of the National Natural Science Foundation of China (NSFC) via grant 62236004 and 62476073.

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

## A  AI TOOLS

In this manuscript, AI tools such as ChatGPT were used solely for refining the text and did not contribute to the content or ideas presented.

## B  HYPER-PARAMETERS

We present the hyperparameter search experiments related to ProxyAttn in this section. Ultimately, we chose a stride of 4 and a minimum budget of 0 for LLaMA and 2048 for Qwen for our main experiments.

### B.1  EFFECT OF GAMMA AND MIN BUDGET

Table 4: The hyperparameter search experiments concerning sparsity on RULER Benchmark.

| $\gamma$ | min budget | sparsity | 4k | 8k | 16k | 32k | 64k | 128k | Avg. | wAvg. |
|---|---|---|---|---|---|---|---|---|---|---|
| | | | | *Llama3.1-8B-Instruct* | | | | | | |
| 0.95 | 0 | 0.69 | 96.07 | 94.57 | 94.01 | 91.55 | 86.63 | 78.25 | **90.18** | **87.43** |
| 0.95 | 1024 | 0.69 | 95.75 | 94.04 | 93.65 | 90.66 | 86.05 | 76.92 | 89.51 | 86.63 |
| 0.95 | 2048 | 0.67 | 95.96 | 94.17 | 93.60 | 90.37 | 85.96 | 75.94 | 89.33 | 86.29 |
| 0.90 | 0 | 0.81 | 95.79 | 94.30 | 93.15 | 90.91 | 85.62 | 76.16 | 89.32 | 86.31 |
| 0.90 | 1024 | 0.80 | 95.82 | 94.23 | 93.31 | 90.18 | 84.72 | 73.63 | 88.65 | 85.25 |
| 0.90 | 2048 | 0.77 | 95.96 | 94.25 | 93.19 | 90.14 | 84.64 | 73.09 | 88.55 | 85.06 |
| | | | | *Qwen2.5-7B-Instruct-1M* | | | | | | |
| 0.95 | 0 | 0.63 | 93.91 | 91.26 | 90.16 | 86.89 | 83.46 | 76.99 | 87.11 | 84.46 |
| 0.95 | 1024 | 0.62 | 94.22 | 90.91 | 90.10 | 87.25 | 83.43 | 77.51 | 87.24 | 84.65 |
| 0.95 | 2048 | 0.61 | 94.61 | 92.07 | 89.75 | 86.68 | 83.57 | 77.09 | **87.30** | **84.53** |
| 0.90 | 0 | 0.76 | 93.59 | 90.27 | 88.34 | 86.19 | 82.10 | 75.84 | 86.06 | 83.31 |
| 0.90 | 1024 | 0.76 | 94.07 | 90.62 | 89.32 | 86.48 | 82.65 | 74.71 | 86.31 | 83.37 |
| 0.90 | 2048 | 0.74 | 94.57 | 91.34 | 89.19 | 86.55 | 83.76 | 76.80 | 87.04 | 84.32 |

### B.2  EFFECT OF STRIDE

As shown in Table 5, a slight increase in the stride does not significantly impair performance due to the local similarity of attention, but it substantially reduces the estimation overhead.

Table 5: The impact of varying strides on the proposed method.

| stride | $\gamma$ | min budget | 4k | 8k | 16k | 32k | 64k | 128k | Avg |
|--------|----------|------------|-----|-----|-----|-----|-----|------|-----|
| *Llama3.1-8B-Instruct* | | | | | | | | | |
| Full Attention | | | 95.63 | 92.27 | 91.60 | 87.66 | 84.85 | 76.17 | 88.03 |
| 1 | | | 95.43 | 92.16 | 90.94 | 89.25 | 84.98 | 74.44 | 87.87 |
| 2 | 0.95 | 0 | 95.32 | 92.13 | 91.46 | 89.29 | 85.24 | 75.56 | 88.17 |
| 4 | | | 95.16 | 92.89 | 91.84 | 89.69 | 84.64 | 76.27 | 88.42 |
| *Qwen2.5-7B-Instruct-1M* | | | | | | | | | |
| Full Attention | | | 93.35 | 89.94 | 87.81 | 87.56 | 85.33 | 76.89 | 86.81 |
| 1 | | | 93.67 | 89.35 | 88.23 | 87.06 | 85.41 | 76.24 | 86.66 |
| 2 | 0.95 | 2048 | 93.67 | 89.88 | 88.54 | 87.66 | 85.13 | 75.10 | 86.66 |
| 4 | | | 93.57 | 90.48 | 88.30 | 86.55 | 84.21 | 74.97 | 86.35 |

## C DETAILED RESULTS ON REAL-WORLD DATASETS

Due to space limitations, the main text reports the average performance across two datasets. The complete experimental results are presented in this section.

Table 6: Main results on InfiniteBench across various models.

| Methods | En | | | | Zh | Code | Math | Retrieval | | | Avg. |
|---------|-----|-----|-----|-----|-----|-------|------|-----------|--------|-----|------|
| | Sum | QA | MC | Dia | QA | Debug | Find | PassKey | Number | KV | |
| *Llama3.1-8B-Instruct* | | | | | | | | | | | |
| FullAttention | 27.12 | 14.37 | 66.38 | 20.50 | 13.65 | 23.35 | 33.43 | 100.00 | 99.32 | 55.20 | 45.33 |
| Minference | 25.51 | 14.30 | 60.70 | 15.50 | 12.57 | 26.40 | 35.71 | 100.00 | 97.97 | 22.80 | 41.15 |
| FlexPrefill | 26.54 | 14.15 | 66.81 | 15.50 | 13.69 | 23.60 | 32.29 | 100.00 | 99.32 | 43.00 | 43.49 |
| XAttention | 26.87 | 14.00 | 69.43 | 15.50 | 13.28 | 22.84 | 32.57 | 100.00 | 99.49 | 40.60 | 43.46 |
| ProxyAttn | 26.78 | 14.85 | 67.69 | 23.50 | 13.35 | 25.13 | 32.00 | 100.00 | 99.66 | 44.80 | **44.78** |
| *Qwen2.5-7B-Instruct-1M* | | | | | | | | | | | |
| FullAttention | 23.48 | 16.86 | 65.94 | 16.00 | 15.65 | 31.98 | 41.14 | 100.00 | 100.00 | 33.60 | 44.47 |
| Minference | 24.37 | 15.87 | 67.25 | 13.00 | 15.16 | 30.20 | 47.71 | 100.00 | 100.00 | 35.60 | **44.92** |
| FlexPrefill | 23.68 | 15.50 | 65.50 | 21.00 | 15.66 | 30.96 | 42.29 | 100.00 | 100.00 | 25.20 | 43.98 |
| XAttention | 22.64 | 15.99 | 66.38 | 13.50 | 15.56 | 32.74 | 43.71 | 100.00 | 100.00 | 11.20 | 42.17 |
| ProxyAttn | 23.16 | 15.26 | 65.07 | 15.50 | 15.36 | 32.99 | 41.71 | 100.00 | 100.00 | 32.80 | 44.19 |

Table 7: Main results on LongBench-v2 across various models.

| Method | Short | | Medium | | Long | | Overall | |
|--------|---------|--------|---------|--------|---------|--------|----------|--------|
| | w.o. Cot | w. Cot | w.o. Cot | w. Cot | w.o. Cot | w. Cot | w.o. Cot | w. Cot |
| *Llama3.1-8B-Instruct* | | | | | | | | |
| FullAttention | 33.90 | 36.10 | 26.50 | 30.70 | 25.00 | 27.80 | 28.80 | 32.00 |
| Minference | 35.00 | 38.90 | 26.00 | 31.20 | 30.60 | 26.90 | 30.20 | **33.00** |
| FlexPrefill | 30.60 | 36.70 | 27.90 | 25.10 | 25.90 | 28.70 | 28.40 | 30.00 |
| XAttention | 35.00 | 35.00 | 26.00 | 31.20 | 26.90 | 27.80 | 29.40 | 31.80 |
| ProxyAttn | 34.40 | 41.10 | 27.40 | 27.90 | 29.60 | 29.60 | 30.40 | 33.00 |
| *Qwen2.5-7B-Instruct-1M* | | | | | | | | |
| FullAttention | 39.40 | 38.90 | 28.40 | 36.30 | 34.30 | 33.30 | 33.60 | 36.60 |
| Minference | 40.60 | 43.90 | 26.50 | 34.40 | 28.70 | 29.60 | 32.00 | 36.80 |
| FlexPrefill | 41.70 | 35.60 | 30.20 | 34.90 | 27.80 | 31.50 | **33.80** | 34.40 |
| XAttention | 41.70 | 42.20 | 27.40 | 33.50 | 32.40 | 30.60 | 33.60 | 36.00 |
| ProxyAttn | 43.90 | 40.00 | 28.40 | 36.70 | 27.80 | 32.40 | **33.80** | **37.00** |

## D SPARSITY RATES

The sparsity rates achieved by different methods using the optimal configuration on RULER. A higher sparsity rate leads to a greater acceleration effect.

Table 8: Sparsity rates achieved by different sparse attention methods across varying input lengths.

| Model | 4K | 8K | 16K | 32K | 64K | 128K |
|---|---|---|---|---|---|---|
| *Llama3.1-8B-Instruct* | | | | | | |
| MInference | 6.15 | 11.62 | 22.60 | 34.13 | 50.44 | 72.24 |
| FlexPrefill | 19.76 | 43.62 | 57.82 | 65.15 | 71.86 | 75.10 |
| XAttention | 29.10 | 39.81 | 49.91 | 58.96 | 68.42 | 73.20 |
| ProxyAttn | **49.63** | **63.21** | **73.31** | **78.27** | **83.19** | **83.86** |
| *Qwen2.5-7B-Instruct-1M* | | | | | | |
| MInference | 25.99 | 32.02 | 39.54 | 50.35 | 61.82 | 72.44 |
| FlexPrefill | 19.39 | **45.02** | 59.26 | 65.50 | 68.39 | 74.35 |
| XAttention | **34.11** | 44.47 | 51.54 | 56.99 | 60.55 | 65.32 |
| ProxyAttn | 13.48 | 42.43 | **60.78** | **70.23** | **74.12** | **79.52** |

# E OBSERVATIONS IN BOTH SYNTHETIC AND REAL-WORLD TASKS

To verify the domain-specific differences of the phenomena observed in Section 2.2, we conducted observational experiments on both synthetic and real-world datasets. As shown in Figures 7 and 8, we observe the cumulative attention score curves for input tokens ranked by both the proxy head and the original heads across different datasets. For synthetic data, we use RULER dataset samples consistent with §2.2, while real-world data are sampled from LongBench v2 (Bai et al., 2024b). It can be observed that the proxy head exhibits relatively consistent behavior across the two different domains, validating the generality of the proposed method.

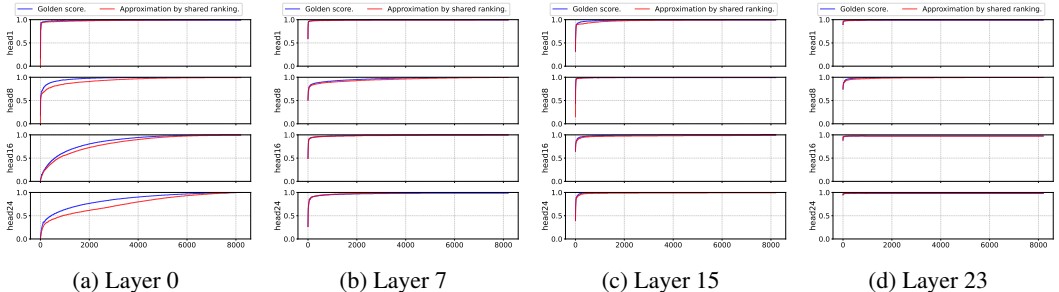

| (a) Layer 0 | (b) Layer 7 | (c) Layer 15 | (d) Layer 23 |

Figure 7: Cumulative attention curves across different layers of Llama3.1-8B-Instruct on synthetic data from RULER.

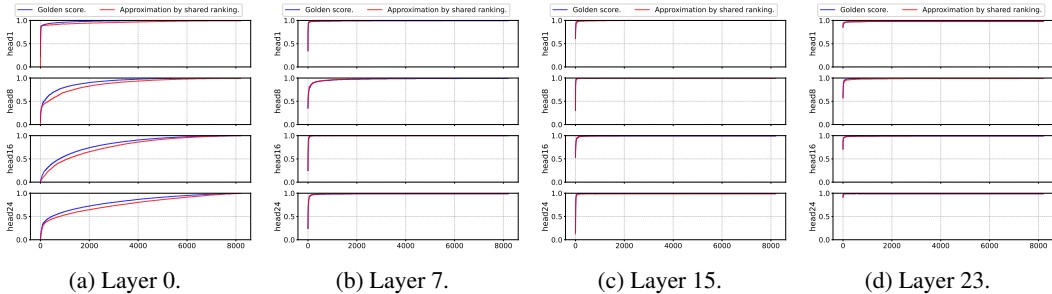

| (a) Layer 0. | (b) Layer 7. | (c) Layer 15. | (d) Layer 23. |

Figure 8: Cumulative attention curves across different layers of Llama3.1-8B-Instruct on real-world data from LongBench v2.

