# OpenReview forum: "ProxyAttn: Guided Sparse Attention via Representative Heads"
_ICLR.cc/2026/Conference — ICLR 2026 Poster_

### Official Review · Reviewer_kQrf · 2025-10-31

**Soundness:** 2
**Presentation:** 3
**Contribution:** 3
**Rating:** 6
**Confidence:** 4

**Summary:**

The paper proposes ProxyAttn, a method for improving block selection in block-sparse attention. Instead of reducing along the sequence dimension to estimate high-attention regions, ProxyAttn computes a subset of attention heads and uses them as proxies to estimate which blocks to compute for the remaining heads. Experiments shows improved performance and efficiency when using the proposed sparse attention method.

**Strengths:**

1. The paper presents an interesting insight into the similarity of attention trends across different heads.
2. The design choices of the proxy attention mask and dynamic budget allocation are well-motivated by corresponding empirical observations and oracle analyses, enhancing the interpretability of the proposed method.
3. The proposed approach outperforms other sparse attention baselines on retrieval tasks (RULER benchmark), matching or even surpassing the dense attention baseline.

**Weaknesses:**

1. **Inconsistent performance on long-context understanding tasks:** Although the proposed method performs well on retrieval tasks, its advantage is less obvious on more realistic long-context benchmarks. This raises questions about whether the observed attention pattern similarity generalizes beyond retrieval settings, since real-world tasks often involve richer semantics and more diverse attention behaviors.

2. **Definition clarity:** Some core concepts, such as *attention score*, are ambiguously defined. For instance, does it refer to the $(i, j)$ entry of the attention matrix, or the aggregated attention value across columns?

3. **Figure clarity:** Several figures are not clearly explained. For example, the meaning of the x-axis in Figure 2(c) is unclear. How does it support the claim that the primary variation between heads lies not in the tokens they attend to?

**Questions:**

1. In the attention head consistency oracle experiment (Figure 2a), how much of the overlap is attributable to attention sinks? What happens if the influence of attention sinks is removed?
2. What does the term “average score” in Line 149 refer to? Is it the mean *attention score* across heads or something else?
3. Does the proxy head mechanism perform consistently across both retrieval and question-answering scenarios?

---

> ### Author Response · Authors · 2025-11-20
> **Response to Reviewer kQrf**
>
> Thank you for your constructive and valuable comments. We provide our responses in the following aspects:
>
> ### **1. The performance of sparse attention methods on real-world QA datasets (regarding Weakness 1, Qeustion 3).**
>
> This is a question well worth investigating. Compared with synthetic tasks, all sparse attention methods exhibit more consistent behavior on real-world tasks.
> We speculate that this phenomenon may arise from the fact that real-world tasks do not sufficiently decouple contextual information from the model’s inherent knowledge.
>
> If the contextual knowledge in a real-world task also appears in the model’s pre-training corpus, the model may ignore the provided context and instead complete the task directly using the knowledge stored in its MLP layers. This can reduce the discriminative power of the final scores, unlike what we observe in synthetic tasks.
> A prior study [1] evaluating KV cache methods provides strong evidence for this point.
> While the StreamLLM approach which retains only local information suffers severe degradation on synthetic tasks (from 99 to 19.3), it incurs only about a 10-point drop on LongBench tasks.
>
> In addition, we provide in Appendix E the cumulative attention score plots obtained using the proxy head on both real-world and synthetic tasks.
> Across all layers, we did not observe any significant differences, which confirms the consistency of the proposed method across data from different domains.
>
> [1] Yuan J, Liu H, Zhong S, et al. Kv cache compression, but what must we give in return? a comprehensive benchmark of long context capable approaches[J]. arXiv preprint arXiv:2407.01527, 2024.
>
> ### **2. Additional clarifications of certain definitions (corresponding to Weakness 2 & 3, Question 2)**
>
> We apologize for the confusion and provide additional clarification regarding the definitions you mentioned.
>
> **Attention score.**
> To facilitate illustrating the interactions between different heads, the attention scores in Figure 2 are taken from the first token in the decoding stage attending to all preceding tokens, with a shape of (head_num, seq_len).
> The explanation of (i,j) in L142 can be formally expressed as follows:
> $$tokenset = topk(A[i], k=1024).indices$$
> $$score = sum(A[j][tokenset])$$
> where tokenset refers to the top 1,024 token indices with the highest attention scores in head i，
> A is the attention score matrix of a given token over all preceding tokens, with dimensions (head_num,seq_len).
>
> **Figure2c.** In Figure 2c, the horizontal axis corresponds to the token indices in the real data, and all four subfigures share the same token ordering. The intended message of Figure 2c is that, although different heads exhibit varying degrees of sparsity, the tokens attended by sparse heads are still those attended by dense heads. In other words, the heads share a common importance ordering over tokens, differing only in how much budget each head allocates.
>
> **Average score in L149.** Your understanding is correct. By “averaging,” we mean taking the mean of the attention scores across different heads (i.e., pooling the QK scores), which is used to distinguish it from the product of pooled QK.
>
> ### **3. Experiments on eliminating the influence of sink tokens (corresponding to Question 1)**
>
> This is an interesting question. We have added in Appendix F the consistency plots across different heads after removing the sink tokens (defined as the first four tokens).
> As the layer depth increases, the attention allocated to sink tokens continues to rise.
> We can see that after removing the sink tokens and normalizing the scores, we still obtain similar conclusions.

---

> > ### Comment · Reviewer_kQrf · 2025-11-27
> >
> > I appreciate the author's revisions and clarifications. The responses address my earlier concerns and clarify the key concepts. Therefore, I am comfortable maintaining my positive score.

---

### Official Review · Reviewer_C449 · 2025-10-31

**Soundness:** 3
**Presentation:** 4
**Contribution:** 2
**Rating:** 6
**Confidence:** 4

**Summary:**

This paper proposes ProxyAttn, an attention approximation method designed to accelerate the pre-filling phase.

The method is based on the key observation that attention patterns across different heads within the same layer are highly similar, differing primarily in their sparsity. Therefore, ProxyAttn employs a pooling head to generate shared attention scores for top-K block retrieval.

Furthermore, the authors propose using the query from each head's final block to estimate its specific sparsity budget, which enables a more effective resource allocation.

**Strengths:**

1.  The paper's motivation is well-articulated. Through multi-faceted comparisons and visualizations, it substantiates the hypothesis that differences between attention heads lie not in their *patterns*, but rather in their *magnitude* and *sparsity*. This insight validates the use of a pooling head for important block selection as a reasonable approach.

2.  The experiments are thorough, conducted across the Ruler, LongBench-v2, and InfiniteBench benchmarks. The evaluation covers two mainstream models, Qwen and LLaMA, lending strong reliability to the results.

**Weaknesses:**

1.  The paper critiques prior methods for coarse-grained block retrieval. However, its own method still employs a stride, resulting in block-wise retrieval and top-k computations. This merely mitigates the coarse-granularity issue rather than fundamentally solving it—a true solution is likely infeasible due to prohibitive retrieval overhead.

2.  The method offers limited speedup, achieving only ~2x on a 70B model. The baseline selection seems insufficient, as current mainstream models often use 7x or 8x GQA ratios. In such high-ratio scenarios, FlashAttention's computational speed increases significantly, representing a more practical setting (however, not optimal for your end-to-end TTFT comparison). Moreover, with the rise of local + global attention models (e.g., Minimax, Llama 4), the significance of pre-filling acceleration is diminishing.

**Questions:**

1. Regarding line 142: "the score at position $(i,j)$ represents the cumulative attention score at head $j$ obtained by the top 1024 tokens from head $i$."This statement is unclear to me. Could you please provide the formula to explain this metric?

2. What is the TTFT of your method on the Qwen2.5-1.5B, 3B, and 7B models?

---

> ### Author Response · Authors · 2025-11-20
> **Response to Reviewer C449**
>
> Thank you for your constructive and valuable comments. We provide our responses in the following aspects:
>
> ### **1. The effect of stride size (corresponding to Weakness 1).**
>
> You raise a crucial point. The concept of stride does indeed somewhat conflict with that of token-level operations. In fact, the existence of stride is primarily aimed at pushing performance to the extreme without compromising overall effectiveness. When stride equals 1, it does not affect the validity of the ProxyAttn itself.
>
> | Method                 | 32k    | 64k    | 128k   | 256k    |
> |------------------------|--------|--------|--------|---------|
> | fullattention          | 22.56  | 90.88  | 416.68 | 1645.27 |
> | minference             | 96.94  | 129.39 | 233.33 | 322.06  |
> | flexprefill            | 18.24  | 41.81  | 119.01 | 248.74  |
> | xattention             | 15.11  | 44.67  | 147.66 | 469.55  |
> | proxyattn (stride=1)   | 12.31  | 30.48  | 83.77  | 202.06  |
> | proxyattn (stride=4)   | 11.45  | 27.17  | 73.07  | 161.60  |
> | proxyattn (stride=8)   | 11.31  | 27.10  | 74.80  | 158.13  |
>
>
> As shown in the table above, we compare the average inference latency of the proposed ProxyAttn under different stride settings in Llama 3.1 (consistent with the configuration used in Figure 4 of the paper).
> Although the latency increases under stride = 1 due to the higher computational cost of estimating attention, the method still maintains a strong advantage compared with other approaches.
> For downstream tasks that demand high performance, we recommend using stride = 1 to achieve genuine token-level estimation.
> In future work, we also plan to explore compressing and quantizing the proxy head dimensions to enable more efficient estimation.
>
> ### **2. Representativeness of the baseline model architecture (corresponding to Weakness 2).**
>
> Through extensive experiments, we find that the GQA ratio is indeed a major factor influencing performance.
> In the paper, we also employ models with diverse GQA ratios (Llama 3.1-8B: 4×, Qwen 2.5-7B: 7×, Llama 3.1-70B: 8×) to validate the effectiveness of the proposed method.
> At least under a purely full-attention architecture, the improvements brought by sparse attention on long documents (exceeding 32K tokens) remain quite substantial.
>
> For some of the new hybrid-attention architectures you mentioned, they indeed pose challenges to existing techniques.
> However, the current lack of support in some inference-framework ecosystems, together with the task-performance degradation observed in such models, is not yet sufficient for them to fully replace existing architectures. I believe that optimization efforts grounded in sparse attention (e.g., the recently released DeepSeek 3.2 DSA) can still provide meaningful gains.
>
> ### **3. Formal clarification of Figure 2a (corresponding to Question 1)**
>
> We apologize for the confusion. The explanation of (i,j) in L142 can be formally expressed as follows:
>
> $$tokenset = topK(A[i], k=1024).indices$$
>
> $$Score[i][j] = sum(A[j][tokenset])$$
>
> where tokenset refers to the top 1,024 token indices with the highest attention scores in head i.
> We assume that A is the attention score matrix of a given token over all preceding tokens, with dimensions (head_num,seq_len).
> In the figure 2a, we use the attention scores of the first token after prefilling with respect to the preceding text.
>
> ### **4. TTFT evaluation of the Qwen series models (corresponding to Question 2)**
>
> We evaluate the TTFT (/s) speedup of our method on larger-scale models using the RULER dataset, as shown in the table below.
>
> | Model                   | Method     | 32K  | 64K  | 128K | 256K |
> |-------------------------|------------|------|------|------|------|
> | qwen2.5-1.5B-Instruct   | flashattn  | 0.59 | 1.48 | 5.16 | 19.53 |
> | qwen2.5-1.5B-Instruct   | proxyattn  | 0.52 | 1.00 | 2.24 | 6.27  |
> | qwen2.5-3B-Instruct     | flashattn  | 1.06 | 2.64 | 9.03 | 32.99 |
> | qwen2.5-3B-Instruct     | proxyattn  | 0.89 | 1.72 | 4.33 | 11.00 |
> | qwen2.5-7B-Instruct     | flashattn  | 1.61 | 4.02 | 13.12 | 46.83 |
> | qwen2.5-7B-Instruct     | proxyattn  | 1.40 | 2.89 | 6.97 | 18.15 |
>
>
> The experimental results are consistent with the patterns we observed on other models.
> We observe approximately a 2× improvement in TTFT at 128k and about a 3× improvement at 256k.

---

### Official Review · Reviewer_N8YJ · 2025-11-01

**Soundness:** 4
**Presentation:** 4
**Contribution:** 4
**Rating:** 8
**Confidence:** 4

**Summary:**

The paper proposes ProxyAttn, a training-free sparse attention algorithm for long-context prefill. The key idea is to compress along the head dimension: group attention heads and use a small number of representative “proxy” heads to compute near token-level scores, then max-pool and aggregate them into fine-grained block importance estimates. To accommodate that different heads prefer different sparsity levels, the method adds a block-aware per-head dynamic budget based on a lightweight cumulative-probability criterion. This yields fine-grained, block-sparse masks at low estimation cost, delivering attention speedups and 2.4× prefill acceleration with no significant performance loss.

**Strengths:**

1. Novel sparse attention design. Instead of compressing along the sequence dimension (tends to be coarse), the paper compresses along the head dimension to obtain token-level importance at the cost of only a few proxy heads, which is conceptually clean and practically effective.
2. No retraining or model changes; works as a drop-in prefill accelerator and ports across model families.
3. Strong speedups while maintaining full-attention accuracy on long context benchmarks.
4.  The approach is validated by the observation that heads share token focus on long context, justifying proxy-head sharing.

**Weaknesses:**

I don't see any substantive weakness.

**Questions:**

1. The method hinges on cross-head token-focus similarity; corner cases with highly specialized heads may degrade the shared ranking quality. Do authors observe tasks or head types where the shared ranking under-represents important blocks (e.g., rare long-range dependencies)?
2. How sensitive is performance to the group size and the choice of the representative head? Could learned or data-driven method further help beyond simple averaging (e.g., fine-tune somes head as proxyheads)?
3. Could the proxy-heads be reused for decoding with query-aware sparse attention?

---

> ### Author Response · Authors · 2025-11-20
> **Response to Reviewer N8YJ**
>
> Thank you for your positive and valuable comments. We provide our responses in the following aspects:
>
> ### **1. The cross-head attention consistency observed in long-context scenarios. (corresponding to Question 1)**
>
> This may seem counterintuitive, but thus far we have not observed strong discrepancies among attention heads in long-context settings, and experimental results across various benchmarks further corroborate this consistency.
> Previous studies on the diversity of attention across different heads have primarily been based on short-text settings (e.g., some heads attending to subjects, others to predicates). However, multiple attention heads receiving the same input essentially constitute a form of ensemble learning, and in very long-context scenarios it is reasonable that they exhibit certain shared ranking patterns.
>
> One possible explanation is that in ultra-long context settings (e.g., beyond 16K tokens), the set of tokens that any given token truly needs to attend to is relatively limited. The differences among heads are therefore concentrated mainly within this small subset, making it feasible to use a proxy head to guide sparse attention.
> We also observe that some prior work [1] performs token pruning directly at the input stage for long-context tasks while still maintaining strong performance, which indirectly supports this conclusion.
>
> [1] Liu J, Chen B, Zhang C. Speculative Prefill: Turbocharging TTFT with Lightweight and Training-Free Token Importance Estimation[J]. arXiv preprint arXiv:2502.02789, 2025.
>
> ### **2. Sensitivity analysis with respect to group size (corresponding to Question 2).**
>
> In Figure 5a, we conduct a group-size analysis on models with both the LLaMA and Qwen architectures.
> As shown, for LLaMA, using a single proxy head shared across all heads already yields strong performance, whereas for Qwen it is preferable to assign one proxy head to each GQA group.
>
> One hypothesis is that the more extreme the GQA configuration, the larger the discrepancies among attention heads across different groups, thus necessitating a separate proxy head for each group.
> Fortunately, under standard tensor-parallel configuration where each GPU holds the attention computation for a single group, assigning an independent proxy head to each group does not introduce significant overhead.
> This also indirectly demonstrates the compatibility of ProxyAttn with modern tensor-parallel and GQA architectures.
>
> ### **3. Other trainable methods for obtaining the ProxyHead (corresponding to Question 2).**
>
> You raise an excellent point, we have also attempted some training in this direction.
> For a proxy head to be effective on long documents, both the length and quality of the training data are highly demanding.
> Under our low-resource training setup, the fine-tuned head do not demonstrate a significant advantage over natural pooling.
> Consequently, to enhance the practicality of the method, we adopt a training-free configuration.
>
> However, it is certain that a proxy head trained with supervised fine-tuning will exhibit performance surpassing that of simple pooling, with the main challenge lying in the training data and scale.
> Excitingly, the indexer in the recently released DeepSeek 3.2 DSA implements a form of trainable proxy head through continued pretraining.
> Its efficient operator implementation and algorithmic design can be seen as partially validating the feasibility of performing sparse attention at the token level.
>
> ### **4. Adaptation of ProxyAttn during the decoding stage (corresponding to Question 3).**
>
> This is an excellent suggestion.
> In fact, we have conducted several experiments on the decoding process, and the behavior of different attention heads during decoding is consistent with the phenomena we observed in the prefilling stage.
>
> However, considering the sparsity computation and operator optimization that differ from those in prefilling, we did not include sparse decoding in the experimental results of this paper for a fair comparison with existing sparse prefilling methods.
> Applying ProxyAttn to the decoding stage to form a unified sparse attention algorithm will be a major direction of our future work.

---

### Official Review · Reviewer_wqai · 2025-11-01

**Soundness:** 3
**Presentation:** 3
**Contribution:** 3
**Rating:** 6
**Confidence:** 3

**Summary:**

The paper introduces ProxyAttn, a training-free sparse attention method designed to speed up long-context Transformers by exploiting redundancy across attention heads. Instead of pruning tokens directly, the authors group similar heads into “proxy heads” and use their aggregated (max-pooled) attention scores to estimate token importance. Each real head then receives its own Top-K allocation based on its intrinsic sparsity. This approach effectively compresses along the head dimension rather than the sequence dimension, which is both novel and impactful. Experiments on large-scale LLMs (LLaMA 3.1, Qwen 2.5) show up to a 10× speedup with almost no drop in accuracy.

**Strengths:**

1.The method can plug into existing Transformer architectures without retraining or parameter tuning, making it practical for deployment.
2.Using proxy heads to estimate token-level importance leads to higher overall sparsity while preserving key contextual information—often matching or slightly outperforming full-attention baselines on long-context benchmarks.
3.The reported efficiency gains are impressive: up to a 10.3× reduction in kernel-level attention time and a 2.4× speedup in total prefilling latency, with accuracy losses staying under one percentage point—an excellent trade-off for latency-critical use cases.

**Weaknesses:**

1.The method is only evaluated during the prefill stage; it’s unclear whether similar benefits (or issues) would arise during autoregressive decoding, especially for long generations.
2.Important hyperparameters—like the sparsity threshold and minimum Top-K—are tuned via grid search. The lack of an adaptive or principled selection rule could make it harder to apply the method out-of-the-box to new models or domains.
3.While computational savings are detailed, the paper doesn’t discuss the memory overhead from storing proxy key–value tensors, which could be a concern on GPUs with limited memory.

**Questions:**

see weakness section.

---

> ### Author Response · Authors · 2025-11-20
> **Response to Reviewer wqai**
>
> Thank you for your constructive and valuable comments. We provide our responses in the following aspects:
>
> ### **1. Adaptation of ProxyAttn during the decoding stage (corresponding to Weakness 1).**
>
> This is an excellent suggestion. In fact, we have conducted several experiments on the decoding process, and the behavior of different attention heads during decoding is consistent with the phenomena we observed in the prefilling stage.
>
> However, considering the sparsity computation and operator optimization that differ from those in prefilling, we did not include sparse decoding in the experimental results of this paper for a fair comparison with existing sparse prefilling methods.
> Applying ProxyAttn to the decoding stage to form a unified sparse attention algorithm will be a major direction of our future work.
>
> ### **2. Hyperparameter selection strategy (corresponding to Weakness 2).**
>
> The grid search conducted in the paper is solely intended to obtain performance results under different sparsity rates, enabling a fair comparison with the optimal settings of the baseline methods.
> Benefiting from the online budget estimation method proposed in this work, ProxyAttn can adaptively select different absolute budgets under varying context lengths for a given $\gamma$ budget (see Table 8 in the appendix).
> For a newly introduced model, we recommend setting γ to 0.95 and using a minimum budget of 1048, which provides a reasonable trade-off between speed and performance.
>
> ### **3. Discussion on the storage overhead of proxy heads (corresponding to Weakness 3).**
>
> We apologize for the confusion caused regarding the storage overhead, and we will provide further clarification in the revised version.
> Since ProxyAttn is applied only during the prefilling stage (each query is processed only once), we do not store any additional intermediate proxy states.
> Consequently, the current implementation of ProxyAttn **does not introduce any substantive storage overhead**.
>
> In future adaptations to the decoding stage, the storage required for proxy heads will become non-negligible, and exploring ways to further compress these states will be highly meaningful.

---

### Public Comment · ~Zewen_Ye1 · 2025-11-13

We have reviewed ProxyAttn with great interest and appreciate its contribution to advancing sparse attention techniques. We would like to draw the authors’ and reviewers’ attention to a prior related work, SharePrefill (https://arxiv.org/abs/2505.19578, May 2025), which appears to share several core ideas with ProxyAttn. Given SharePrefill’s temporal precedence and the conceptual similarities between the two works, in particular:

- Both studies observe consistent similarity across attention heads,  as stated in SharePrefill: “(1) attention patterns demonstrate strong inter-head similarity, and (2) this similarity remains remarkably consistent across diverse inputs” and in ProxyAttn: “attention heads exhibit consistency in token focus.”

- SharePrefill’s approach of  "(1) Offline clustering to group heads based on the similarity of their attention score maps. (2) Online inference, where pivotal attention is constructed dynamically and shared with other heads during the inference process." appears conceptually related to ProxyAttn’s method of “using scores from pooled representative heads to approximate those of all heads.”

We believe that explicitly clarifying the distinctions between ProxyAttn and SharePrefill would further strengthen the paper by highlighting its unique contributions within the existing body of work. We would be happy to provide more detailed technical comparisons if helpful.

---

> ### Author Response · Authors · 2025-11-20
>
> Thank you for your interest in our work. After reading the SharePrefill paper you mentioned, I agree that it is a constructive and valuable contribution.
> However, I must emphasize that it differs from ProxyAttn in several fundamental ways, the two most fundamental differences can be summarized as follows:
>
> **1. Conceptual differences between sharing and proxying.**
>
> Although both methods emphasize consistency, SharePrefill focuses on exact equivalence, leading to direct mask sharing across heads.
> ProxyAttn, in contrast, observes that heads share a similar ranking of tokens but differ in sparsity, and therefore uses a proxy head to estimate a shared sequence while assigning different budgets to each head.
> This means that the proxy head does not enforce complete sharing; instead, each head maintains its own specific mask.
>
> **2. Differences in the granularity of block-importance estimation.**
>
> In my view, the key factor behind ProxyAttn’s performance improvement is token-level importance estimation, with proxy heads serving merely as an efficient implementation of this estimation.
> By compressing along the head dimension, ProxyAttn can efficiently obtain token-level importance scores at low cost.
> In contrast, SharePrefill appears to perform conventional compression along the sequence dimension, which may overlook a small number of high-scoring tokens.
>
> We apologize for omitting the discussion of SharePrefill in the related work section and will include it in the revised version.
> However, as noted above, SharePrefill and ProxyAttn are **fundamentally two distinct sparse attention approaches**, each with its own characteristics.
> Once the SharePrefill code is released, we are willing to include corresponding comparative experiments.

---

> > ### Public Comment · ~Zewen_Ye1 · 2025-11-24
> >
> > Thank you for the careful review of SharePrefill and the comparison with ProxyAttn. I would like to clarify one point to ensure the comparison is accurate.
> >
> > In “2. Differences in the granularity of block-importance estimation,” you mentioned that ProxyAttn performs token-level importance estimation, whereas SharePrefill does not, stating that “SharePrefill appears to perform conventional compression along the sequence dimension, which may overlook a small number of high-scoring tokens.”
> >
> > In fact, the pivotal attention in SharePrefill is exact full attention without any compression. As stated in the SharePrefill paper, it “computes the full attention for a subset of heads to identify the actual sparse patterns,” and “the head computes full attention using a dense pattern (i.e., a pattern with all ones).” Therefore, the comment about compression does not apply. From the perspective of granularity, SharePrefill also operates at the token level.
> >
> > I hope this clarification is helpful for improving the related work comparison.

---

> > > ### Author Response · Authors · 2025-11-25
> > >
> > > Thank you for your correction. After a closer look at SharePrefill, I summarize the technical differences of ProxyAttn as follows:
> > > 1. **Estimation method**: Unlike SharePrefill, which estimates using a single key head, ProxyAttn performs estimation using a pooled proxy head.
> > > 2. **Budget allocation**: Observing differences in sparsity across heads, ProxyAttn applies distinct dynamic budgets to different heads.
> > > 3. **Implementation approach**: SharePrefill still employs a hybrid of VS and Block patterns, with VS dominating, whereas ProxyAttn uses a purely block sparse attention method.
> > >
> > > We have added a discussion of SharePrefill in the Related Work section. We hope that both works can inspire further research on architecture and algorithm design guided by inter-head similarity.

---

### Comment · Area_Chair_xdRE · 2025-11-23
**Reviewer & Author Discussion**

Hi Reviewers,

Please kinly and actively participate in the review-author dicussion, raise your further concerns so that the authors can explain more, and make your final decisions.

---

### Meta-Review · Area_Chair_7FCe · 2026-01-05

**Summary:**

### Strength emphasized by reviewers
* Well-motivated.
* Provide an interesting and well-supported insight about 'differences between attention heads lie not in their patterns, but rather in their magnitude and sparsity.
* Novel method design.
* Design choices are sound and solid with empirical observation and oracle analyses.
* Experiments are thorough.
* Practical for deployment.
* The method outperforms baselines on retrieval tasks, matching or even surpassing the dense attention baseline.

### Ensemble of concerns and suggestions (well addressed by the author rebuttal)
1. About clarity
    * Some concepts are ambiguously defined [Reviewer kQrf].
    * Some figures are not clearly explained [Reviewer kQrf].
2. About the method design
  	* The method still employs a stride, resulting in block-wise retrieval, and thus is somewhat conflict with the advocated token-level importance estimation [Reviewer C449].
  	* Need an adaptive or principled selection rule to avoid grid search on the hyperparameters (e.g., sparsity threshold) [Reviewer wqai].
3. About experimental results
    * The advantage on realistic long-context benchmark is relatively small, raising question about whether the observed attention pattern similarity generalize beyond retrieval settings [Reviewer kQrf].
    * The method offers limited speedup, achieving only 2x on a 70B model [Reviewer C449]. And as the work only targets the prefilling stage, it's not clear whether similar benefits would arise during decoding stage [Reviewer wqai].

**Reviewer Concerns:**

All concerns are addressed by the author rebuttal. In some concerns, the reviewer proposed good and reasonable suggestions, but those directions could be addressed in future work (e.g., implementation in the decoding stage, adaptive sparsity selection method).

**Reviewer Scores:**

I think all reviewer will at least keep their positive scores.

---

### Decision · Program_Chairs · 2026-01-26

Accept (Poster)